# communications
# earth & environment

# Wildfire imagery reduces risk information-seeking among homeowners as property wildfire risk increases

Hilary Byerly Flint [1,2✉], Patricia A. Champ[3], James R. Meldrum [4] & Hannah Brenkert-Smith[1]

Negative imagery of destruction may induce or inhibit action to reduce risks from climate-exacerbated hazards, such as wildfires. This has generated conflicting assumptions among experts who communicate with homeowners: half of surveyed wildfire practitioners perceive a lack of expert agreement about the effect of negative imagery (a burning house) on homeowner behavior, yet most believe negative imagery is more engaging. We tested whether this expectation matched homeowner response in the United States. In an online experiment, homeowners who viewed negative imagery reported more negative emotions but the same behavioral intentions compared to those who viewed status-quo landscape photos. In a pre-registered field experiment, homeowners who received a postcard showing negative imagery were equally likely, overall, to visit a wildfire risk webpage as those whose postcard showed a status quo photo. However, the negative imagery decreased webpage visits as homeowners' wildfire risk increased. These results illustrate the importance of testing assumptions to encourage behavioral adaptation to climate change.

[1] Institute of Behavioral Science, University of Colorado Boulder, Boulder, CO 80309, USA. [2] Haub School of Environment and Natural Resources, University of Wyoming, Laramie, WY 82072, USA. [3] Rocky Mountain Research Station, U.S.D.A. Forest Service, Fort Collins, CO 80526, USA. [4] U.S. Geological Survey, Fort Collins Science Center, Fort Collins, CO 80526, USA. ✉email: hflint1@uwyo.edu

Climate change is causing more frequent and intense natural hazards, including wildfires, floods, and droughts[1-3]. These events generate powerful negative imagery of destruction and loss, which is often publicized in the media (e.g., refs. [4-6]). Practitioners—employees of government agencies and nonprofits working to encourage and support individual actions to mitigate and adapt to these hazards—must decide whether to use such negative imagery in their risk communications.

Visually highlighting negative consequences of climate-induced hazards could cause individuals to act for several reasons. Dramatic photos can increase information-seeking and pro-social behavior[7,8]. To the extent that a photo elicits negative emotions, such as fear and worry, these emotions can be a "wellspring of action,"[9] motivating people to reduce risk[10]. Negative emotions can motivate behavior by intensifying attention[11] and making threats seem more proximate (i.e., reducing psychological distance)[12]. Correlational studies have consistently found negative emotions to be associated with climate change mitigation and adaptation behavior[13,14].

People also tend to be loss-averse, such that avoiding losses is preferred to seeking equivalent gains[15]. Negative emotions are associated with loss framing[16], and negatively framing outcomes has increased favorable attitudes towards disaster preparedness[17]. Moreover, making possible consequences salient can influence judgments about how likely those consequences are to occur[18]. Imagery of flooding leads to higher perceived risks of flooding[19], and nearby past fire events are associated with reduced housing prices and increased government funding for fuels reduction projects[20,21].

Fear appeals, which highlight the severity and saliency of threats, are a common strategy in risk communications[22,23]. According to Protection Motivation Theory, the effects of fear appeals depend on cognitive appraisal processes, including perceptions of severity, vulnerability, and response efficacy and costs[24]. Adaptive responses to fear appeals are guided by coping appraisal, which includes ability to address the threat (self-efficacy), effectiveness of action (response efficacy), and response costs. When perceptions of efficacy are high, fear appeals can increase protective behaviors[22,25]. This pathway has been established empirically in the health communications literature, but few empirical studies have causally linked appeals and action in climate and environmental domains[26].

Conversely, highlighting the negative consequences of climate risks could dissuade action. Maladaptive responses to fear appeals (e.g., avoidance or denial) may occur when there are intrinsic or extrinsic rewards for avoiding the threat[24], or when perceptions of high severity and vulnerability are not accompanied by efficacy (according to the Extended Parallel Process Model)[27]. While imagery of climate impacts increases threat salience, it rarely elicits self-efficacy[28]. Without self-efficacy, negative emotions can generate hopelessness[29]. Undesirable emotions or information could prompt willful ignorance, avoidance, or even reactance among those who do not want to be scared into submission[30]. There is also a risk of compassion fatigue, in which emotional appeals lead to exhaustion or desensitization[31], or skepticism, if emotional appeals or strong terminology reduce a message's credibility[32,33].

Uncertainty about the effects of negative framing and emotions has led to controversy over their use in climate change communications[34]. The terms "climate crisis" and "climate emergency" are increasingly used by news outlets[33], and scientists have called for greater use of images that make the negative effects of climate change more salient[35]. Yet theories and empirical work from other domains suggest this approach may backfire if it is not accompanied by effective ways to take action[26]. Claims about behavioral effects of fear appeals and negative emotions in climate communications are often supported by evidence that is correlational, from experiments with small samples, or lacking domain specificity[13,34,36]. As a result, there is an "evidence-advice" disparity about the use of negative imagery that may have produced confusion among practitioners[36]. Given the challenges associated with isolating the elements of any given image, experimental research should seek ecological validity aligned with the needs of practitioners[37].

Here, we investigate practitioner assumptions about the use of negative imagery in risk communication and the effect of negative imagery on behavioral adaptation to a climate-exacerbated hazard in the United States. We conducted three studies using a photo of a burning house that was published in *National Geographic* alongside a story about wildfires in California[38]. With the photographer's permission, this image served as the treatment (*flames photo*) to identify the effects of negative imagery in communications with homeowners in areas at risk from wildfire. Images of wildfire destruction and smoke can evoke concern for climate change[39]. The comparison was an aerial photo of a wildfire-prone community (*status quo photo*; Fig. 1), representing an approach currently used by wildfire practitioners in homeowner communications (further justification provided in Methods).

In the first study, we surveyed wildfire practitioners who work with homeowners to encourage risk mitigation on private properties ($n = 120$) and asked them to predict whether the flames or status quo photo would be more effective in engaging homeowners to learn more about their properties' wildfire risk. We also asked which photo they preferred in risk communications, as well as whether they perceive agreement within their field about the use of flames imagery to influence homeowner behavior.

In an online experiment, homeowners ($n = 440$) in wildfire-prone states responded to flames and status quo photos, allowing us to identify relevant differences between the two images described above. Homeowners were told to imagine they live in an area at risk from wildfire and that they had received a photo postcard from the local fire department. They were randomly

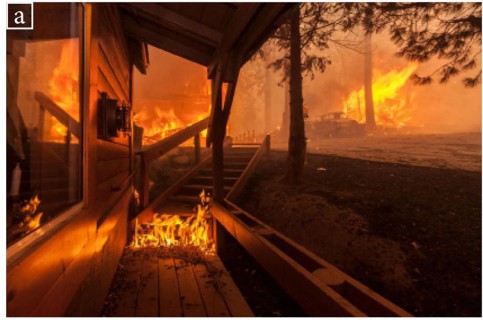 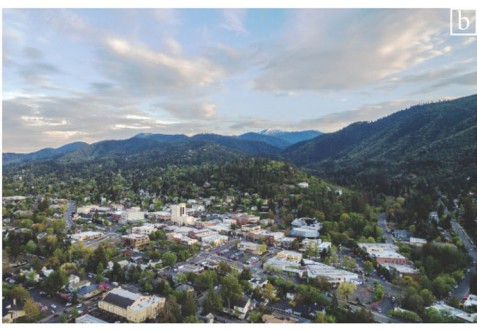

**Fig. 1 Photos used across the three studies. a** Flames photo (Photo credit: Mark Thiessen). **b** Status quo landscape photo (Photo credit: Darren Campbell).

assigned to one of four photos: the flames photo, the status quo photo, or one of two similar photos as part of a sensitivity analysis (Supplementary Fig. 1). After viewing the assigned photo, participants were asked about emotional reactions, probability assessments, behavioral intentions, and personal preferences. We tested for differences between the two photos used across the three studies (Fig. 1), and report those results below. As a sensitivity analysis, we also tested whether differences are consistent with similar wildfire photos used in mainstream media and homeowner communications. Those results are provided in the Supplementary Information.

Finally, in a field experiment in a wildfire-prone city in the American West, we tested the effect of the flames photo on homeowners' information-seeking behaviors. The field experiment was conducted in Ashland, Oregon, which is designated as within a Wildfire Hazard Zone[40], and in collaboration with the wildfire division of the local fire department. In March 2018 the wildfire division assessed the wildfire risk of every property in the city, scoring points for each risk factor (defensible space, structure characteristics, emergency access, and background fuels). These data were used to populate property-specific wildfire risk webpages for each property in the city (Supplementary Fig. 2).

In July 2020 the wildfire division launched a campaign to increase wildfire risk awareness and mitigation by mailing postcards to homeowners (Supplementary Fig. 3) and directing them to visit their property-specific webpages, providing residents with the first opportunity to view their risk information. Homeowners ($n = 5785$) were stratified based on their properties' wildfire risk and randomly assigned to receive one of two versions of the postcard, which varied by the image on the front (Fig. 1). Half received the flames photo of the burning house. The other half received the status quo photo of the local community. Descriptive statistics of the sample and balance across treatments are shown in Supplementary Table 1. Webpage visits were tracked at the homeowner level using access codes unique to each household, which were entered to view a property's webpage.

We also tested whether response to the flames photo varied by property wildfire risk. A previous study using a similar design found that residents that own higher-risk properties were more likely to seek mitigation information when they were informed of their properties' wildfire risk[41]. The Risk Information Seeking and Processing Model also predicts that higher risk leads to information insufficiency and more information-seeking, particularly when negative emotions are high[42,43]. We pre-registered our hypotheses, methods, and analysis plan (see https://osf.io/fa2ms).

## Results

**Wildfire practitioner survey**. In our survey of wildfire practitioners, 83% of respondents believed the negative imagery of flames is more effective in engaging homeowners to learn about their wildfire risk (Table 1; $\chi^2 = 135.4$, $p < 0.001$). Most respondents (62%) would also choose to use the flames photo in their communications ($\chi^2 = 47.5$, $p < 0.001$). However, 29% of those who chose the flames photo as most effective would not choose to use that photo in their communications. In explaining their choice, nine respondents (31%) used fear-related terms (see Methods). For example, one practitioner wrote, "[the flames photo] could put folks off who perceive it as a scare tactic." Overall, 29% of respondents used fear-related terms in justifying which photo they believed to be most effective.

Regarding perceived agreement about using negative imagery in risk communications within their field, 48% responded that most practitioners think flames photos engage homeowners, while 43% believed there is no agreement about the effect of flames photos (Table 1). Only 3% of respondents indicated that

most practitioners do not think photos matter in communicating with homeowners.

**Online experiment**. Homeowners who viewed the flames photo reported feeling different emotions than those who viewed the status quo photo (Fig. 2). Compared to the status quo photo, the flames photo caused participants to feel more anxious (difference in means ($M_{\mathrm{diff}}$) = 2.3 percentage points, 95% confidence interval (CI) = 1.7 to 3.0), fearful ($M_{\mathrm{diff}} = 2.0$, 95% CI = 1.3 to 2.6), and worried ($M_{\mathrm{diff}} = 2.3$, 95% CI = 1.7 to 2.9), and less calm ($M_{\mathrm{diff}} = -1.8$, 95% CI = $-2.4$ to $-1.1$), peaceful ($M_{\mathrm{diff}} = -2.1$, 95% CI = $-2.8$ to $-1.5$) and safe ($M_{\mathrm{diff}} = -2.2$, 95% CI = $-2.8$ to $-1.6$). Test statistics and $p$-values are reported in Supplementary Table 2.

The flames photo also caused respondents to rate their imagined home as slightly riskier than those who viewed the status quo photo ($M_{\mathrm{diff}} = 0.8$, 95% CI = 0.2 to 1.4), however respondents in both groups assessed the probability of wildfire and chance of damages about the same. There was also no difference in behavioral intentions after viewing the two photos. When asked, "If the local fire department created a website that shows wildfire risk information specific to your property, how likely are you to visit this website?" those who viewed either photo responded they were, on average, "somewhat likely" to visit the website ($M_{\mathrm{diff}} = 0.0$, 95% CI = $-0.1$ to 0.2). There were differences in personal responses to the two photos: the flames photo was less liked ($M_{\mathrm{diff}} = -3.0$, 95% CI = $-3.7$ to $-2.3$) and deemed less personally relevant ($M_{\mathrm{diff}} = -1.7$, 95% CI = $-2.5$ to $-0.9$) than the status quo photo.

These differences were consistent with responses to a different flames photo used in mainstream media and a different status quo photo used by another wildfire organization (Supplementary Table 3 and Supplementary Fig. 4).

**Field experiment**. Overall, 19.3% of homeowners who received postcards visited their property-specific wildfire risk webpages. Fewer homeowners visited their webpages after receiving the flames photo compared to those who received the status quo photo, but this difference is not statistically significant (Flames = 18.8%, Status Quo = 19.7%; $\chi^2$ (1, $n = 5785$) = 0.7, $p = 0.4$; Supplementary Table 4). The regression-adjusted difference in webpage visits between the two photos is similar ($\beta = -0.7$; $p = 0.5$; 95% CI [$-2.7$, 1.3]; Table 2). The regression model includes adjustments for parcel and owner characteristics, including property acreage and value.

Regardless of the photo on their postcard, homeowners with higher-risk properties were more likely than those with low-risk properties to visit their personalized risk webpages. Response to the negative imagery, however, changed with property wildfire risk (Supplementary Fig. 5). Controlling for this interaction, webpage visits among homeowners with a wildfire risk score of zero were higher for those who received the flames photo than the status quo photo ($\beta = 9.1$ percentage points; $p < 0.01$; 95% CI [2.7, 15.5]; Table 2). However, for every 100-point increase in risk score—which ranged from zero to 1000 points—the flames photo decreased web visits by 2.2 percentage points compared to the status quo photo ($p < 0.01$; 95% CI [$-3.8$, $-0.8$]; Fig. 3). For a homeowner whose risk score is one standard deviation higher than average, the flames photo reduced webpage visits by 3.5 percentage points compared to the status quo baseline. These results are robust to model specification without adjustments and the exclusion of very high and very low-risk score values (Table 2).

## Discussion

Wildfire practitioners we surveyed overwhelmingly expected negative imagery to be more effective in engaging homeowners to

**Table 1 Responses to wildfire practitioner survey.**

| | | More preferred photo | | | |
|---|---|---|---|---|---|
| | | **Flames** | **Status Quo** | **Neither** | **Total** |
| More effective photo | Flames | 60% (71) | 6% (7) | 17% (22) | 83% (100) |
| | Status Quo | 1% (2) | 7% (8) | 3% (3) | 11% (13) |
| | No difference | 1% (2) | 2% (2) | 3% (3) | 6% (7) |
| | Total | 62% (75) | 15% (17) | 23% (28) | |
| Is there agreement between wildfire practitioners about whether photos of flames (i.e., worst-case scenario) engage or repel homeowners when communicating about wildfire risk? | | | | | |
| Yes, most practitioners think flames engage homeowners | | | | | 48% (58) |
| Yes, most practitioners think flames repel homeowners | | | | | 3% (4) |
| No, there is not agreement about the effect of flames photos | | | | | 43% (52) |
| Most practitioners do not think photos matter in homeowner communications | | | | | 3% (4) |

*Note:* Participants (n = 120) indicated whether the flames photo was more effective or preferred than the status quo photo in communications with homeowners (top) and their perceptions of agreement about the use of flames imagery in communications (bottom). Cells show proportion (N).

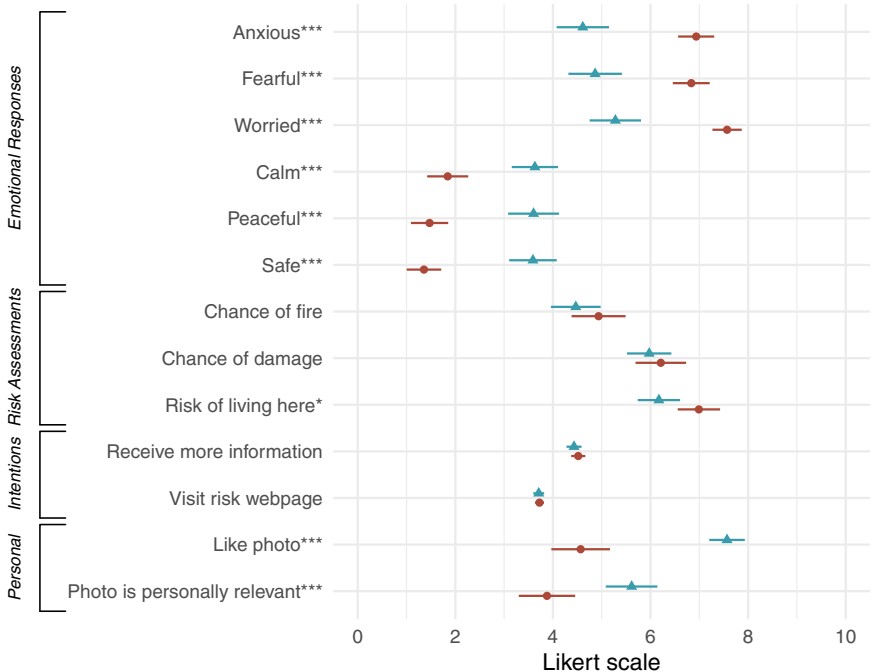

**Fig. 2 Differences between self-reported responses to the flames photo and status quo photo in the online experiment.** Red circles represent mean responses to the flames photo; blue triangles represent mean responses to the status quo photo. Lines show ±1 standard error of the mean. Emotional responses and personal preferences were rated on a Likert scale from "Not at all" (0) to "The most possible" (10). Risk assessments were rated from "No chance" (0%) to "For sure" (100%) and "Not risky at all" (0) to "Extremely risky" (10). Behavioral intentions were rated on a 5-point Likert scale from "Extremely unlikely" (1) to "Extremely likely" (5) and doubled for this figure. p-values are from t-tests and adjusted for multiple comparisons using the Benjamini–Hochberg method. *p < 0.05; ***p < 0.001.

seek additional information about their wildfire risk. When tested, neither the behavioral intentions nor the observed behavior of homeowners supported this expectation. In fact, among higher-risk properties—where mitigation is most important—the flames photo reduced homeowner engagement compared to a status quo landscape image. Nearly a third of the practitioners who believed the flames photo to be more effective would not use it in their communications with homeowners, perhaps signaling awareness of the controversy around negative imagery and its potential to induce a maladaptive response.

The flames photo induced stronger negative emotions and weaker positive emotions than the status quo photo among homeowners in the online experiment. If these emotional effects replicated in the field, perhaps fear and worry attracted the attention of all who received the flames photo, but those emotions

interacted with the personal wildfire risk information on the postcard. Fear appeals tend to work best when accompanied by high perceived efficacy, or effective and feasible options for responding to the threat[44]. Receiving the flames photo and then learning one's property was low risk may have encouraged a sense of efficacy, enabling protective information-seeking behavior[45]. Conversely, homeowners who received the flames photo and then learned their property was high risk were faced with more challenging coping strategies. This combination of high susceptibility to and low efficacy to avert a threat can lead people to engage in fear control responses[27]. These include avoidance (not visiting the webpage to avoid further difficult thoughts and feelings), willful ignorance (choosing to remain uninformed to avoid making undesirable decisions), reactance (considering the message "fear mongering") or even hopelessness. A study on COVID-19

**Table 2 The estimated average treatment effect of a flames photo on homeowners' information-seeking behavior in the field experiment.**

| | Average treatment effect (ATE) | | Conditional ATE: risk score | | Conditional ATE excluding outliers | |
|---|---|---|---|---|---|---|
| Model | 1 | 2 | 3 | 4 | 5 | 6 |
| Flames Photo | −0.009 | −0.007 | 0.084* | 0.091** | 0.096** | 0.090** |
| | (0.010) | (0.010) | (0.033) | (0.033) | (0.036) | (0.034) |
| Risk Score/100 | 0.038** | 0.037** | 0.048** | 0.048** | 0.051** | 0.048** |
| | (0.004) | (0.004) | (0.005) | (0.005) | (0.006) | (0.005) |
| Flames photo × risk score/100 | — | — | −0.021** | −0.022** | −0.023** | −0.022** |
| | | | (0.008) | (0.008) | (0.008) | (0.008) |
| Ownership Characteristics | No | Yes | No | Yes | Yes | Yes |
| Parcel Characteristics | No | Yes | No | Yes | Yes | Yes |
| Observations | 5785 | 5701 | 5785 | 5701 | 5571 | 5618 |

Note: The estimated effect is the difference in webpage visitation (proportion) between the flames photo and the status quo photo. Linear probability estimation with robust standard errors. The estimated moderating effect of Risk Score is for a 100-point change in a parcel's wildfire risk score. Models 1 and 2 show the main effects of the flames photo without and with household-level covariates. Models 3 and 4 show the effects of the flames photo conditional on parcel-level risk score. Results in the manuscript are reported from Model 4. Additional specifications exclude possible outliers of risk score (very low and very high scores) using the Interquartile Range criterion (Model 5), or observations greater than the 97.5th percentile or less than the 2.5th percentile (Model 6). Ownership characteristics include part-time ownership and owning multiple properties. Parcel characteristics include wildfire risk score, year built, acreage, and value. Standard errors in parentheses. — omitted from the regression. *Statistical significance at the 5% level; **statistical significance at the 1% level.

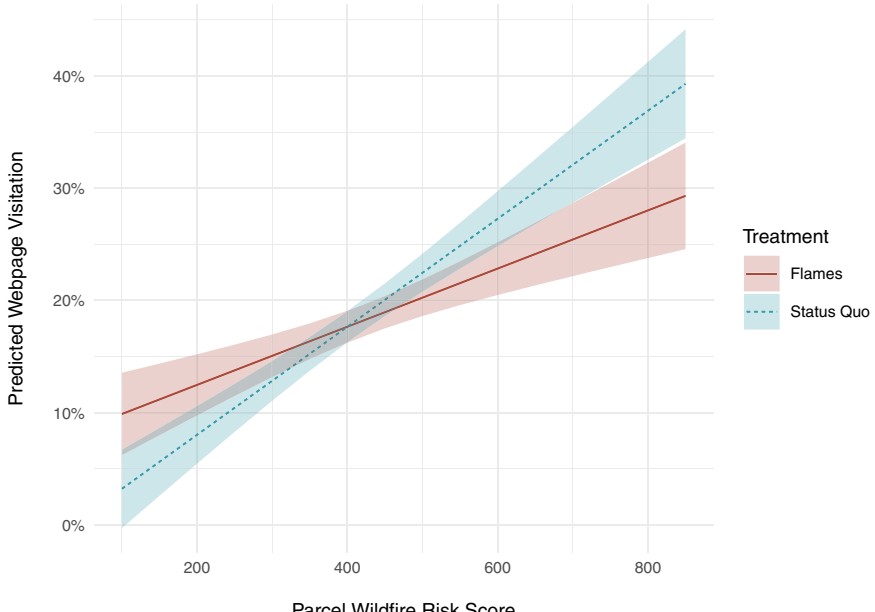

**Fig. 3 Predicted webpage visitation by property-level wildfire risk and photo treatment.** Solid red line represents predicted visitation in response to the flames photo; dashed blue line represents predicted visitation in response to the status quo photo. Shaded ribbons represent the 95% confidence intervals for the predicted values.

protective behaviors found that increased risk perceptions about infectiousness were associated with decreased willingness to engage in social distancing—a "fatalism effect"[46]. If such undesirable responses occurred among high-risk homeowners in our study, we consider a deeper question: is their failure to mitigate risk on their property linked to the same feelings and responses that were elicited when that risk was made salient through worst-case-scenario imagery?

The importance of perceived efficacy in response to the flames imagery is consistent with both the wildfire context[47] and the broader theoretical literature on fear appeals and negative affect in climate communications[27,48]. Wildfire risk to private property includes factors that are beyond a homeowner's control[49]. A densely forested neighboring property and a steep hillside can greatly increase a property's wildfire risk, yet there is little the homeowner can do to change those factors. In this case, perceived efficacy is likely to be low and the homeowner

will attempt to reduce fear rather than danger[27]. Even factors within a homeowner's control, such as roof material or defensible space, can be costly to change and even cost-prohibitive for those who are resource-constrained[50]. The response costs or "resource-related attributes" of mitigation action (time, financial cost, effort, and cooperation requirements) may have induced a maladaptive response[24,48]. That the flames imagery increased webpage visitation among low-risk property owners but reduced visitation among high-risk property owners may precisely demonstrate the "tradeoff between salience and efficacy" of fear appeals[37], and the "threat-by-efficacy" interaction explained by the Extended Parallel Process Model[27]. Moreover, while climate change may be perceived as so large and unsolvable as to inhibit self-efficacy and response efficacy[26], highlighting even more localized climate-related threats (e.g., to one's own property) may be paralyzing enough to induce maladaptive responses.

However, it is unknown whether the photos in the field experiment elicited the same emotional responses as in the online experiment. Laboratory experiments, in which subjects are aware of their participation in research, are subject to biases in responses, including from perceived expectations of researchers (*experimenter demand effect*) or efforts to present oneself in a positive manner (*social desirability bias*). The scrutiny, context, and subject pool selection in the online experiment may have influenced whether those effects are generalizable to the field[51]. It is also unknown whether homeowners in different communities would respond similarly. Ashland is a relatively dense wildland-urban interface city with a median housing value more than double that of the United States and a population that is older, whiter, and more educated than national population averages[52].

Another possible explanation for the field experiment results is related to the "risk perception gap" between the public and experts. Homeowners consistently rate their wildfire risk lower than do wildfire professionals[53]. The professionally determined risk score on the postcard may have generated cognitive dissonance if high-risk homeowners believed their wildfire risk to be lower. This disconnect, combined with the negative imagery, may have led to threat denial[32] or psychological reactance[54]. Future research might parse out these and other potential explanations for the negative response to negative imagery among high-risk homeowners, as well as effects on different outcomes that matter for wildfire risk-reduction and over longer time frames.

We note that the wildfire professionals who responded to our survey may not represent the broader population of wildfire practitioners. These individuals are part of an online community of fire professionals interested in building community resilience to wildfire. As such, these practitioners may be more informed about and experienced with wildfire outreach to homeowners. It is unclear whether or how their perceptions might differ from the broader population of wildfire practitioners. However, this sub-population is integral to building fire-adapted communities in the United States[55].

The key result of the field experiment for these and other practitioners is that negative imagery and fear appeals can backfire among those who are at greatest risk to climate-related hazards. Related theory suggests the importance of providing feasible options for risk-reduction to avoid this negative response[27,48]. While the postcard in the field experiment advertised a website with "specific information on how to reduce wildfire risk," this text may not have been enough to attenuate the resource-related or efficacy concerns of high-risk homeowners. Identifying and addressing those concerns may be key to using negative imagery in risk communications, which may indeed have attention-grabbing benefits.

Many wildfire practitioners believe negative imagery matters in risk communications with homeowners. While the overall behavioral response in this study contradicts this belief, a closer look shows that negative imagery matters differently for different homeowners. As practitioners are encouraged to pursue evidence-based policies with input from the social and behavioral sciences[56], collecting data on and testing for these moderating factors is critical to evaluating assumptions and informing effective outreach, particularly for vulnerable subgroups. Collaborations between researchers and practitioners can improve efforts to promote risk mitigation as threats from wildfire, hurricanes, and other natural disasters are projected to grow in a changing climate.

## Methods

**Study 1: Wildfire practitioner survey**. We used a convenience sample of wildfire practitioners (individuals who work for non-governmental organizations and government agencies on wildfire issues). Practitioners were recruited through emails to contacts and postings on the Fire Adapted Communities Learning Network (https://fireadaptednetwork.org/) message board. This is a national network of wildfire professionals actively working on fire adaptation, including representatives from fire departments, conservation districts, nonprofits, and Firewise fire councils. In order to detect whether the proportion of practitioners who believe the flame photo is at least 10 percentage points more or less effective compared to the status quo photo ($H_0$: $p = 0.5$; $H_A$: $0.4 \leq p \geq 0.6$), we continued surveying until we received at least 100 responses. Ass we do not know how many of the network's members viewed the posting, we are unable to measure the response rate.

Using an online survey (Supplementary Methods), practitioners were shown the flames photo and the status quo photo (Fig. 1) and then asked, "Which photo is more effective at engaging homeowners to learn more about their wildfire risk?" and "Which photo would you choose to use in your outreach communications to homeowners about wildfire risk? For example, on a postcard informing homeowners about their properties' wildfire risk." We also asked why practitioners selected a particular photo (open text) and about perceived agreement within their field regarding whether photos of flames (i.e., worst-case scenario) engage or repel homeowners. Proportions were calculated from responses, and differences were tested using Pearson's chi-squared tests. Qualitative text responses regarding why a photo was selected were coded as including fear-related terms if they included any of the following: "fear," "scare," "scary," "panic," "danger," "devastating," "doom," "anxiety," "shock," "alarmist," "dramatic," "emotional," "worst-case scenario." The selection of these terms was based on participant responses.

**Study 2: Online experiment**. The sample included online research participants from Prolific (www.prolific.co) who are self-reported homeowners and reside in one of the 15 most wildfire-prone states in the United States[57]. Participants were compensated one dollar for completing the questionnaire, which took about five minutes.

The introduction to the questionnaire asked participants to imagine they own a home in an area at risk from wildfire and there have been 10 large wildfires in the area in the last 100 years. Participants were then told they had received a postcard from the local fire department stating their property is "high risk" for wildfire and showing a photo on the other side. The photo shown to a participant was randomly assigned from one of four photos (Supplementary Fig. 1). Two photos represented the status quo because they had been used by fire departments to communicate with homeowners about wildfire risk. Two photos represented flames imagery because they showed burning houses and had appeared in media outlets along with stories about wildfire destruction. Participants were then asked questions about emotional reactions, risk assessments of wildfire, intentions to seek information about wildfire risk, and personal preferences regarding the two photos. Study design and questions were modeled after similar studies on disaster risk perceptions and emotional reactions to imagery[11,19,58,59]. Specifically, participants were asked to imagine they owned a home in an area at risk from wildfire to make the hazard relevant and provide a specific, consistent risk measure for all participants[19]. Responses were provided on Likert scales that were defined according to the question but represented the range from "not at all" (0) to "the most possible" (5 or 10). The full questionnaire is provided in Supplementary Methods.

The goals of this study were (1) to measure differences between the two photos used in the field experiment; and (2) to determine whether reactions to those two photos generalized to two similar photos. Differences between the field experiment photos (Goal 1) were evaluated using t-tests with Benjamini–Hochberg corrections for multiple comparisons[60]. Generalizability of similarities and differences to the other two photos (Goal 2) was tested using one-way analyses of variance and post hoc Tukey's Honestly Significant Difference test. Ethical approval was provided by the corresponding author's Institutional Review Board (Protocol # 20-0372 approved 7/23/20).

A similar study design, which showed participants photos of flooding and asked about risk perceptions, found effect sizes ranging from 0.35 to 0.63[19]. A meta-analysis on the effects of fear appeals found an average weighted effect size of $d = 0.29$[22]. Based on these results, the present study sought to detect an effect of $d = 0.35$. A power analysis determined a minimum sample of about 100 participants in each of the four treatments at $\alpha = 0.05$ and $\beta = 0.80$. We estimated that up to 10% of responses might be incomplete, thus a final sample size of 440.

**Study 3: Field experiment**. The sample frame included owners of every parcel within the city limits of Ashland, Oregon ($N = 6400$). Property owners were identified using the Jackson County Property Data list (https://web.jacksoncounty.org/pdo/). Non-residential properties were excluded from the list. Owners of multiple properties were identified using a name-matching process, which resulted in 5785 unique postcard recipients. Homeowners were also identified as part-time residents if their mailing address was outside the city. A mailing list was constructed from the Property Data list and matched to parcel-level wildfire risk assessment data collected by the local wildfire division.

Postcards were mailed to all homeowners in Ashland, Oregon by the wildfire division of the local fire department in July 2020. The postcard informed recipients of their parcel's wildfire risk and directed them to visit a personalized webpage to learn more about their risk factors and resources for taking action (Supplementary

Figs. 2 and 3). This was the first effort by the wildfire division to inform homeowners of their wildfire risk score, although they had sent previous wildfire-related mailings to some homeowners. Homeowners were randomly assigned to receive one of two versions of the postcard (Fig. 1): either the treatment, which had an image of a burning house (flames photo), or the control, which had an aerial image of the local landscape (status quo photo). The status quo photo is commonly used by Ashland's local government in wildfire communications, including on two website homepages, a previous postcard mailing, and a flyer about homebuying and wildfire safety. Assignment to treatment followed a randomized block design, in which homeowners were randomly and evenly distributed between the treatment and control groups according to their wildfire risk rating (Low to Extreme), which was classified by wildfire experts using wildfire risk scores. A unique code on the postcard allowed the recipient to access their webpage. Webpage visits were tracked for the two months following the mailing using Google Analytics.

The outcome of interest was whether a household visited their personalized wildfire risk webpage. To test for a difference between webpage visits among those who received the flames photo and those who received the status quo photo, we used a Pearson's chi-squared test. Then we estimated a linear probability model with robust standard errors without and with household-level covariates, including parcel wildfire risk score, parcel acreage, property value, and whether the owner's mailing address is outside the city or owns multiple properties in the city (Table 2; Models 1 and 2, respectively). The moderator analysis included an interaction between the treatment and parcel risk score (Table 2; Models 3 and 4). We also tested whether results were sensitive to very high and very low-risk scores by excluding those observations according to the Interquartile Range criterion (Table 2; Models 5) and excluding observations greater than the 97.5th percentile or less than the 2.5th percentile (Table 2; Model 6). For ease of interpretability, parcel risk score was divided by 100 in all models. Both the main and moderator analyses were pre-registered. Ethical approval was provided by the University of Colorado, Boulder Institutional Review Board (Protocol # 20-0177 approved 4/16/20), including a waiver of informed consent.

**Reporting summary**. Further information on research design is available in the Nature Research Reporting Summary linked to this article.

## Data availability
The data that support the findings of this study are available on the project site on Open Science Framework (https://osf.io/fa2ms), with the exception of potentially identifying parcel characteristics from Study 3. The practitioner and online surveys are included in the supplementary information.

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

## Acknowledgements
We are tremendously grateful to Katie Gibble and Chris Chambers at Ashland Fire and Rescue for making the field experiment possible. We thank Chris Barth and Jamie Gomez for their expert input, and Colleen Donovan and Carolyn Wagner for data and logistical support. This research was supported by National Science Foundation (NSF) Grant SES-1823509, United States Department of Agriculture Forest Service Rocky Mountain Research Station, the NSF- and Federal Emergency Management Agency-funded Mitigation Matters Research Program through the Natural Hazards Center, Institute of Behavioral Science Small Grants Program, and Ashland Fire and Rescue. The survey described in this report was organized and implemented by the University of Colorado Boulder and was not conducted on behalf of the U.S. Geological Survey. The findings and conclusions in this paper are those of the authors and should not be construed to represent any official USDA or U.S. Government determination or policy.

## Author contributions
H.B.F., P.C., J.M., and H.B.-S. designed research; H.B.F. performed research; H.B.F. analyzed data; H.B.F., P.C., J.M., and H.B.-S. wrote the paper.

## Competing interests
The authors declare no competing interests.
