## [Peer Review File · Communications Earth & Environment]

Web links to the author's journal account have been redacted from the decision letters as indicated to maintain confidentiality.

24th Jan 22

Dear Dr Byerly,

Please allow me to apologise for the long delay in sending a decision on your manuscript titled "The effects of wildfire imagery in risk communications". It has now been seen by 3 reviewers, and I include their comments at the end of this message. They find your work of interest, but some important points are raised. We are interested in the possibility of publishing your study in Communications Earth & Environment, but would like to consider your responses to these concerns and assess a revised manuscript before we make a final decision on publication.

We therefore invite you to revise and resubmit your manuscript, along with a point-by-point response that takes into account the points raised. Please highlight all changes in the manuscript text file.

In particular, please ensure that in the revised manuscript you:

- * Clearly explain the aspects of your methodology that are highlighted as potentially confusing or contradictory by the reviewers.
- * Place your findings firmly in the context of existing relevant literature and theory.

Please use the following link to submit your revised manuscript, point-by-point response to the referees' comments (which should be in a separate document to any cover letter) and the completed checklist:

[link redacted]

We hope to receive your revised paper within six weeks; please let us know if you aren't able to submit it within this time so that we can discuss how best to proceed. If we don't hear from you, and the revision process takes significantly longer, we may close your file. In this event, we will still be happy to reconsider your paper at a later date, as long as nothing similar has been accepted for publication at Communications Earth & Environment or published elsewhere in the meantime.

We understand that due to the current global situation, the time required for revision may be longer than usual. We would appreciate it if you could keep us informed about an estimated timescale for resubmission, to facilitate our planning. Of course, if you are unable to estimate, we are happy to accommodate necessary extensions nevertheless.

Please do not hesitate to contact me if you have any questions or would like to discuss these revisions further. We look forward to seeing the revised manuscript and thank you for the opportunity to review your work.

Best regards,

Joe Aslin

Senior Editor,
Communications Earth & Environment
<https://www.nature.com/commsenv/>
Twitter: @CommsEarth

EDITORIAL POLICIES AND FORMATTING

Editorial Policy: [Policy requirements](https://www.nature.com/documents/nr-editorial-policy-checklist.zip)

Furthermore, please align your manuscript with our format requirements, which are summarized on the following checklist:

[Communications Earth & Environment formatting checklist](https://www.nature.com/documents/commsj-phys-style-formatting-checklist-article.pdf)

and also in our style and formatting guide [Communications Earth & Environment formatting guide](https://www.nature.com/documents/commsj-phys-style-formatting-guide-accept.pdf) .

***** DATA:** Communications Earth & Environment endorses the principles of the Enabling FAIR data project (<http://www.copdess.org/enabling-fair-data-project/>). We ask authors to make the data that support their conclusions available in permanent, publically accessible data repositories. (Please contact the editor if you are unable to make your data available).

All Communications Earth & Environment manuscripts must include a section titled "Data Availability" at the end of the Methods section or main text (if no Methods). More information on this policy, is available at <http://www.nature.com/authors/policies/data/data-availability-statements-data-citations.pdf>.

DATA SOURCES: All new data associated with the paper should be placed in a persistent repository where they can be freely and enduringly accessed. We recommend submitting the data to discipline-specific, community-recognized repositories, where possible and a list of recommended repositories

is provided at <http://www.nature.com/sdata/policies/repositories>.

If a community resource is unavailable, data can be submitted to generalist repositories such as [figshare](https://figshare.com/) or [Dryad Digital Repository](http://datadryad.org/). Please provide a unique identifier for the data (for example a DOI or a permanent URL) in the data availability statement, if possible. If the repository does not provide identifiers, we encourage authors to supply the search terms that will return the data. For data that have been obtained from publically available sources, please provide a URL and the specific data product name in the data availability statement. Data with a DOI should be further cited in the methods reference section.

REVIEWER COMMENTS:

Reviewer #1 (Remarks to the Author):

The aim of this paper is to examine whether imagery of climate-related disasters (specifically wildfires) motivates people to adapt to such risks, a question that remains currently unresolved in the literature on climate change communication. I think that this paper is likely of interest to a large audience of both academics as well as practitioners. The combination of different types of studies is excellent, and the field experiment especially is a great example of how adaptation behaviour can be assessed in the field. It also seems to me that the statistical analyses are done carefully, but I couldn't find a link to the preregistration or data of the studies so I wasn't able to check this in detail.

The main thing that I think can be improved with this paper is its connection to the previous literature on this topic. While there is a short discussion of some relevant constructs such as fear appeals and self-efficacy, I find that this section does not really do justice to the extensive literature that is already available on the use of fear appeals. Specifically, I am thinking of a more extensive discussion of the theoretical mechanisms behind fear appeals, such as protection motivation theory, which explains the conditions under which fear appeals should or should not work. The role of negative affect in motivating behaviour could also be explained better. I think that if this paper was better embedded in the existing literature, the findings reported here would be relevant not only to practitioners designing wildfire prevention campaigns, but would also contribute valuable insights to the evidence base on fear appeals in general.

I think explaining previous theorizing more extensively in the introduction may also help the authors to interpret their own findings better. For example, the authors clearly find that the fear appeal had its intended effect in Study 2 (increasing negative affect), but that this in turn did not influence behavioral intentions. In Study 3, they find that the effect of the images differed across assigned risk levels. The authors do offer some explanation in the discussion for these findings, mainly referring to levels of efficacy, but I think the discussion of these findings could be tied more strongly to existing theoretical frameworks and their associated findings. It is not quite clear now how the current study

relates to previous work in the literature. Does it support or subvert theories and previous findings in the literature? What new theoretical insights were gained from this study, especially since the study relied on experimental field data? In turn, this could also make the practical implications of this study stronger; how can practitioners overcome this emotion-behaviour gap?

In a similar vein, I think this paper can be connected better to the ongoing discourse on the use of fear appeals in climate change communication specifically. The authors hint to this literature in their introduction, but I think a more thorough treatment of this discussion is appropriate here. There is currently quite the discussion raging about whether fear appeals should even be used in climate communications at all, even though there is a steadily growing body of literature that suggests that negative affect is (at least in correlational studies) a consistent and strong motivator of mitigation and adaptation behaviour (for a partial overview of this debate, see (Ettinger, Walton, Painter, & DiBlasi, 2021). Moreover, outside of academia, terminology such as climate crisis and climate emergency are increasingly used (Feldman & Hart, 2021), and some scientists have urged news outlets to also use imagery that reflects the severity of climate change (see for example <https://www.carbonbrief.org/guest-post-how-heatwave-images-in-the-media-can-better-represent-climate-risks>). I am missing a discussion of and references to papers from this discussion in the current paper (such as (Bloodhart, Swim, & Diccico, 2019; Brosch, 2021; Duan, Hepworth, Ormerod, & Canon, 2021; Ettinger et al., 2021; Feldman & Hart, 2021; Kuller, Schoenholzer, & Lienert, 2021; O'Neill & Smith, 2013; Reser & Bradley, 2017). An overview of this discussion can also be used to better justify Study 1, since this discussion in the literature at large has likely also affected how practitioners think about the use of negative imagery in practice. I am also interested in reading how the authors interpret their own findings in light of this discussion. Do the findings reported in this paper generally support the use of negative-affective images or not?

Minor point

Line 151 -153: But for every 100-point increase in risk score, the flames photo decreased web visits by 2.3 percentage points (Figure 4; $p < 0.01$; 95% CI [-3.8, -0.8]).

To help the interpretation of this statement, it would be useful to provide the scale of the risk score (e.g., does it range from 0 to 1000?).

References

- Bloodhart, B., Swim, J. K., & Diccico, E. (2019). "Be Worried, be VERY Worried:" Preferences for and Impacts of Negative Emotional Climate Change Communication. *Frontiers in Communication*, 3. doi:10.3389/fcomm.2018.00063
- Brosch, T. (2021). Affect and emotions as drivers of climate change perception and action: a review. *Current Opinion in Behavioral Sciences*, 42, 15-21. doi:10.1016/j.cobeha.2021.02.001
- Duan, R., Hepworth, K. J., Ormerod, K. J., & Canon, C. (2021). Promoting Concern for Climate Change: A Study of Wildfire Photographs Using Q Methodology. *Science Communication*, 43(5), 624-650. doi:10.1177/107554702111041689
- Ettinger, J., Walton, P., Painter, J., & DiBlasi, T. (2021). Climate of hope or doom and gloom? Testing the climate change hope vs. fear communications debate through online videos. *Climatic Change*, 164(1-2). doi:10.1007/s10584-021-02975-8
- Feldman, L., & Hart, P. S. (2021). Upping the ante? The effects of "emergency" and "crisis" framing in climate change news. *Climatic Change*, 169(1-2). doi:10.1007/s10584-021-03219-5
- Kuller, M., Schoenholzer, K., & Lienert, J. (2021). Creating effective flood warnings: A framework from a critical review. *Journal of Hydrology*, 602. doi:10.1016/j.jhydrol.2021.126708

O'Neill, S. J., & Smith, N. (2013). Climate change and visual imagery. *WIREs Climate Change*, 5(1), 73-87. doi:10.1002/wcc.249

Reser, J. P., & Bradley, G. L. (2017). Fear Appeals in Climate Change Communication. In *Oxford Research Encyclopedia of Climate Science*.

Reviewer #2 (Remarks to the Author):

This paper presents the findings of a well-designed, timely and relevant study of the effects of imagery in communication about wildfire risks to homeowners. The combination of an expert survey, an online (laboratory) experiment, and a field experiment is commendable and provides a rich source of internally and externally valid results. The findings shed light on the interaction effects between use of imagery in risk communication and the actual reported risk in numerical terms on information-seeking behaviour by homeowners. The novel – and practically very relevant – finding is that risk imagery (here: a wildfire photo) tends to depress information-seeking behaviour the higher the actual reported risk is to a homeowner, compared to more “peaceful” imagery. This, of course, has implications for the usefulness of impact imagery in risk-related communication in times of a climate crisis (whether that’s public outreach campaigns, media coverage, corporate communication, etc.).

I’d like to see a version of this paper in print, since it contributes to the burgeoning field of visual effects research in climate change communication, and because of its usefulness to practitioners in climate risk communication. I have a few points of critique which relate to mainly three areas: (a) the role and significance of the expert survey; (b) the use of and reporting of confidence intervals for non-randomly selected data; and (c) the exact design of the online experiment. Plus, there are some minor editorial questions.

Please make sure that the geographical (and therefore political, cultural, economic, etc.) scope of your study is made explicit right from the start. It was the expression “wildfire-prone states” (line 69) that made me realize this is a U.S.-based study (actually, I already thought so because of the lack of specification in the abstract and opening paragraphs); please introduce the U.S. focus in the abstract.

As regards the expert survey, I can see how it helps to illustrate the gap between practitioners’ perceptions and behavioural inclinations on the one hand and your experiments’ findings on the other. But beyond that it seems rather peripheral to and detached from the overall message of the paper.

Relatedly, you report confidence intervals for your descriptive findings from the survey. Since these results are not based on a randomly drawn sample, the CI numbers don’t make a lot of sense. I’d also like to know a bit more about the message board you used for recruitment: Is it nationwide? How well does it match the population of “wildfire practitioners in the U.S.?” Do you have a notion of what kind of biases have been introduced by your recruitment and sampling strategy? What was the response rate?

I’m getting a bit confused about the design of the online experiment and the respective findings. Figure 1 only show two images and you often speak of the “flames photo” and the “status-quo photo” (singular) when describing the findings of the online experiment. But later on, you mention

four photos and your supplementary material shows the other two images. You randomly showed four photos to your participants (Table S2), but I'm not sure what the reported findings of the online experiment in the main text are referring to. Are the calculations just based on the experimental exposure to the images in Figure 1 (i.e., the ones also used in the field experiment), or are they based on all four treatment conditions (but grouped into two)? I assume the later, but it's just a bit confusing – especially when you write (on lines 259-260) that you wanted “to determine whether reactions to those two photos generalized to two similar photos.” Lines 137-183 suggest that you ran the tests twice.

Lastly, you say (on lines 88-89) that “homeowners were randomly assigned based on their properties' wildfire risk” (repeated on lines 290-291). That sounds contradictory. Were they assigned randomly or based on a pre-set criterion? I assume that you first grouped households based on their risk scores (e.g., everyone with a score of 700 was assigned to one group) and then within each risk-level-group the stimulus material was assigned randomly to households. That'd make sense. Can you please describe it so that nobody gets confused?

The online questionnaire could be added to the supplementary material.

The following are some minor editorial comments:

- line 25: maybe explain in a few words who these practitioners are (as you do later in the manuscript);
- line 76: where exactly did you find these photos and why did you choose them as experimental stimuli?
- line 94: you could be more specific and write “unique access codes for every household, which needed to be entered when visiting the website” or something similar;
- line 116: should read “no agreement;”
- line 147: some readers might wonder what you mean by “parcel” here; you could also list two or three owner characteristics for illustrative purposes;
- line 166: “did induce more negative and less positive emotions” is correct but sounds a bit clunky.

Dr Antal Wozniak (University of Liverpool, Department of Communication and Media)

Reviewer #3 (Remarks to the Author):

Thank you for the opportunity to review “The effects of wildfire imagery in risk communications”. In this manuscript, the authors explore evocative negative imagery and wildfire risk communication. The authors survey wildfire practitioners on their attitudes about the use of negative images, and conduct two surveys of members of the public to assess how negative images influences willingness to engage in information seeking about protective behavior. Overall, I think the authors did a good job and the study will be particularly useful for practitioners looking to design risk communication campaigns informed by best available science. The statistical analyses are appropriate for the research questions proposed by the authors and the nature of the data collected. I have one issue with interpretation of the results, as described in greater detail below. The authors provide suitable detail to reproduce this study, although they may want to consider including their survey items as supplementary material in the appendix.

I enjoyed reading the paper and I hope the authors find these comments helpful.

Major Comments

1. Lines 98 – 99: “Theory also predicts...” Practitioners and academic readers of Communications Earth & Environment may be unfamiliar with many of the theories of risk communication frequently utilized in the homeowner preparedness/wildfire risk communication literature. The authors may want to consider whether they should name and define some theories and models of risk communication and risk appraisal (e.g., Risk Information Seeking and Processing Model, Extended Parallel Process Model, Protection Motivation Theory, etc.) both here and more generally in the introduction and discussion. This may help readers situate the results in a larger theoretical context.
2. The authors find that wildfire practitioners typically believe the evocative negative imagery is more effective at motivating information-seeking behavior, the effect of negative imagery depends on the characteristics of the individual. The authors recommend researchers and practitioners collaborate to improve risk communication to encourage mitigation behavior. I think the paper would be strengthened by a discussion of management implications to advise practitioners on when and how to use negative imagery, given the literature the authors reviewed in the introduction, the results of this study, and the potential theoretical justifications the authors explore in the discussion.
3. Some parts of the online experiment are unclear. From my understanding, the authors asked respondents to imagine they lived in a home at risk of wildfire. Were respondents asked how likely they would be to visit a personalized website for their imagined residence or for their actual residence? Can the authors provide some background on why they chose to have respondents imagine they owned a home at risk of wildfire rather than have respondents answer from the perspective of their current residence?
4. I’m not sure I follow some of the interpretation of the field experiment in the discussion. Lines 151 – 157, the authors mention twice that seeing the flames photo makes high-risk respondents less likely to visit the webpage. I would clarify that for high risk respondents, the negative emotions photo made them less likely to visit the webpage than high risk respondents who received the neutral photo. Based on Figure 4, it looks like respondents at high fire risk were more likely to view the webpage than low risk respondents regardless of treatment. However, the authors seem to imply in lines 170 – 180 that those who were highly susceptible to wildfire risk may be more likely to engage in fear control responses like avoidance and willful ignorance (even though across both conditions, the higher your susceptibility, the more likely you were to seek out information).

Methods and Figures

1. How did the authors ensure their respondents were homeowners? In the Ashland, OR, sample, how did the authors treat residences where the occupants were renters? Did the authors exclude multi-family structures like apartments in addition to non-residential properties?
2. Did the authors send multiple waves of postcards in the field experiment, or just one?
3. Were any of the parcel characteristics or ownership characteristics related to information-seeking behavior?
4. Table S1 should include the descriptive statistics for the sub-sample who visited the webpage. I recommend replacing the Flames and Status Quo columns with two columns for the Flames subsample who visited the page and the Status Quo subsample who visited the page. I recommend Table S1 be brought into the main manuscript.
5. Figure 2: I recommend the authors replace Figure 2 with Table S2.
6. Figure 3: I recommend the authors replace Figure 3 with Table S3.

Minor Comments

1. The authors include the reference number superscript inside the period rather than outside throughout the manuscript.
2. In the supplementary materials, the figures and tables are not listed in the order they appear in the manuscript. For example, the authors reference Figure S3 and S4 on line 138, well before Figures S1 and S2 are references on line 283.
3. Extremely minor, but consider adding in the title of Figure S1 that the example address is the Ashland Fire and Rescue and not a home address of a potential respondent.

Reviewer #1 (Remarks to the Author):

The aim of this paper is to examine whether imagery of climate-related disasters (specifically wildfires) motivates people to adapt to such risks, a question that remains currently unresolved in the literature on climate change communication. I think that this paper is likely of interest to a large audience of both academics as well as practitioners. The combination of different types of studies is excellent, and the field experiment especially is a great example of how adaptation behaviour can be assessed in the field. It also seems to me that the statistical analyses are done carefully, but I couldn't find a link to the preregistration or data of the studies so I wasn't able to check this in detail.

We thank the reviewer for providing helpful comments on our manuscript and sharing references to literature that we had overlooked. These comments and references have greatly improved how we position our study in the literature, theory, and current debate around fear appeals in climate change communication. They have also enabled us to better understand and discuss our results, hopefully providing more clarity to the reader on the importance of efficacy in response to negative imagery. We describe these revisions in the point-by-point responses to the reviewer's comments below.

The main thing that I think can be improved with this paper is its connection to the previous literature on this topic. While there is a short discussion of some relevant constructs such as fear appeals and self-efficacy, I find that this section does not really do justice to the extensive literature that is already available on the use of fear appeals. Specifically, I am thinking of a more extensive discussion of the theoretical mechanisms behind fear appeals, such as protection motivation theory, which explains the conditions under which fear appeals should or should not work. The role of negative affect in motivating behaviour could also be explained better. I think that if this paper was better embedded in the existing literature, the findings reported here would be relevant not only to practitioners designing wildfire prevention campaigns, but would also contribute valuable insights to the evidence base on fear appeals in general.

We have made considerable revisions to the introduction of our paper to better connect it to the previous literature. These revisions include a discussion of the theoretical mechanisms behind fear appeals, including protection motivation theory, and the conditions under which fear appeals should or should not work (lines 46-66):

“Fear appeals, which highlight the severity and saliency of threats, are a common strategy in risk communications.^{22,23} According to Protection Motivation Theory, the effects of fear appeals depend on cognitive appraisal processes, including perceptions of severity, vulnerability, and response efficacy and costs.²⁴ Adaptive responses to fear appeals are guided by coping appraisal, which includes ability to address the threat (*self-efficacy*), effectiveness of action (*response efficacy*), and response costs. When perceptions of efficacy are high, fear appeals can increase protective behaviors.^{22,25} This pathway has been established empirically in the health communications literature, but few empirical studies have causally linked appeals and action in climate and environmental domains.²⁶

Conversely, highlighting the negative consequences of climate risks could dissuade action. Maladaptive responses to fear appeals (e.g., avoidance or denial) may occur when there are intrinsic or extrinsic rewards for avoiding the threat,²⁴

or when perceptions of high severity and vulnerability are not accompanied by efficacy (according to the Extended Parallel Process Model).²⁷ While imagery of climate impacts increases threat salience, it rarely elicits self-efficacy.²⁸ Without self-efficacy, negative emotions can generate hopelessness.²⁹ Undesirable emotions or information could prompt willful ignorance, avoidance, or even reactance among those who do not want to be scared into submission.³⁰ There is also a risk of compassion fatigue, in which emotional appeals lead to exhaustion or desensitization,³¹ or skepticism, if emotional appeals or strong terminology reduce a message's credibility.^{32,33}

We have also added an explanation of the role of negative affect in motivating behavior (lines 34-37):

“Negative emotions can motivate behavior by intensifying attention¹¹ and making threats seem more proximate (i.e., reducing psychological distance).¹² Correlational studies have consistently found negative emotions to be associated with climate change mitigation and adaptation behavior.^{13,14}”

I think explaining previous theorizing more extensively in the introduction may also help the authors to interpret their own findings better. For example, the authors clearly find that the fear appeal had its intended effect in Study 2 (increasing negative affect), but that this in turn did not influence behavioral intentions. In Study 3, they find that the effect of the images differed across assigned risk levels. The authors do offer some explanation in the discussion for these findings, mainly referring to levels of efficacy, but I think the discussion of these findings could be tied more strongly to existing theoretical frameworks and their associated findings. It is not quite clear now how the current study relates to previous work in the literature. Does it support or subvert theories and previous findings in the literature? What new theoretical insights were gained from this study, especially since the study relied on experimental field data? In turn, this could also make the practical implications of this study stronger; how can practitioners overcome this emotion-behaviour gap?

We have revised the discussion to address these comments. First, in a paragraph that interprets our results in the context of the existing theoretical literature (lines 225-241):

“The importance of perceived efficacy in response to the flames imagery is consistent with both the wildfire context⁴⁷ and the broader theoretical literature on fear appeals and negative affect in climate communications.^{27,48} Wildfire risk to private property includes factors that are beyond a homeowner's control.⁴⁹ A densely forested neighboring property and a steep hillside can greatly increase a property's wildfire risk, yet there is little the homeowner can do to change those factors. In this case, perceived efficacy is likely to be low and the homeowner will attempt to reduce fear rather than danger.²⁷ Even factors within a homeowner's control, such as roof material or defensible space, can be costly to change and even cost-prohibitive for those who are resource-constrained.⁵⁰ The response costs or “resource-related attributes” of mitigation action (time, financial cost, effort, and cooperation requirements) may have induced a maladaptive response.^{24,48} That the flames imagery increased webpage visitation among low-risk property owners but reduced visitation among high-risk property owners may precisely demonstrate the “tradeoff between salience and efficacy” of fear appeals,³⁷ and

the “threat-by-efficacy” interaction explained by the Extended Parallel Process Model.²⁷ Moreover, while climate change may be perceived as so large and unsolvable as to inhibit self-efficacy and response efficacy,²⁶ highlighting even more localized climate-related threats (e.g., to one’s own property) may be paralyzing enough to induce maladaptive responses.”

Then, in a paragraph that discusses the practical significance of our results (lines 271-279):

“The key result of the field experiment for these and other practitioners is that negative imagery and fear appeals can backfire among those who are at greatest risk to climate-related hazards. Related theory suggests the importance of providing feasible options for risk-reduction to avoid this negative response.^{27,48} While the postcard in the field experiment advertised a website with “specific information on how to reduce wildfire risk,” this text may not have been enough to attenuate the resource-related or efficacy concerns of high-risk homeowners. Identifying and addressing those concerns may be key to using negative imagery in risk communications, which may indeed have attention-grabbing benefits.”

In a similar vein, I think this paper can be connected better to the ongoing discourse on the use of fear appeals in climate change communication specifically. The authors hint to this literature in their introduction, but I think a more thorough treatment of this discussion is appropriate here. There is currently quite the discussion raging about whether fear appeals should even be used in climate communications at all, even though there is a steadily growing body of literature that suggests that negative affect is (at least in correlational studies) a consistent and strong motivator of mitigation and adaptation behaviour (for a partial overview of this debate, see (Ettinger, Walton, Painter, & DiBlasi, 2021). Moreover, outside of academia, terminology such as climate crisis and climate emergency are increasingly used (Feldman & Hart, 2021), and some scientists have urged news outlets to also use imagery that reflects the severity of climate change (see for example <https://www.carbonbrief.org/guest-post-how-heatwave-images-in-the-media-can-better-represent-climate-risks>). I am missing a discussion of and references to papers from this discussion in the current paper (such as (Bloodhart, Swim, & Diccico, 2019; Brosch, 2021; Duan, Hepworth, Ormerod, & Canon, 2021; Ettinger et al., 2021; Feldman & Hart, 2021; Kuller, Schoenholzer, & Lienert, 2021; O'Neill & Smith, 2013; Reser & Bradley, 2017). An overview of this discussion can also be used to better justify Study 1, since this discussion in the literature at large has likely also affected how practitioners think about the use of negative imagery in practice. I am also interested in reading how the authors interpret their own findings in light of this discussion. Do the findings reported in this paper generally support the use of negative-affective images or not?

We thank the reviewer for these helpful suggestions and references, which have been incorporated into the paper. We have elaborated on the current debate around fear appeals in climate change communication in the introduction and the likely implications for practitioners (lines 67-79):

“Uncertainty about the effects of negative framing and emotions has led to controversy over their use in climate change communications.³⁴ The terms “climate crisis” and “climate emergency” are increasingly used by news outlets,³³

and scientists have called for greater use of images that make the negative effects of climate change more salient.³⁵ Yet theories and empirical work from other domains suggest this approach may backfire if is not accompanied by effective ways to take action.²⁶ Claims about behavioral effects of fear appeals and negative emotions in climate communications are often supported by evidence that is correlational, from experiments with small samples, or lacking domain specificity.^{13,34,36} As a result, there is an “evidence-advice” disparity about the use of negative imagery that may have produced confusion among practitioners.³⁶ Given the challenges associated with isolating the elements of any given image, experimental research should seek ecological validity aligned with the needs of practitioners.^{37”}

We also now note the relevance of this controversy to the results of our practitioner survey (lines 199-202):

“Nearly a third of the practitioners who believed the flames photo to be more effective would not use it in their communications with homeowners, perhaps signaling awareness of the controversy around negative imagery and its potential to induce a maladaptive response.”

The added discussion of the practical significance of our results elaborates on whether the findings support the use of negative affective images (lines 271-279).

Minor point

Line 151 -153: But for every 100-point increase in risk score, the flames photo decreased web visits by 2.3 percentage points (Figure 4; $p < 0.01$; 95% CI [-3.8, -0.8]). To help the interpretation of this statement, it would be useful to provide the scale of the risk score (e.g., does it range from 0 to 1000?).

We have clarified this in the text (lines 184-187):

“But for every 100-point increase in risk score—which ranged from zero to 1000 points—the flames photo decreased web visits by 2.3 percentage points compared to the status quo photo”

References

- Bloodhart, B., Swim, J. K., & Diccico, E. (2019). “Be Worried, be VERY Worried:” Preferences for and Impacts of Negative Emotional Climate Change Communication. *Frontiers in Communication*, 3. doi:10.3389/fcomm.2018.00063
- Brosch, T. (2021). Affect and emotions as drivers of climate change perception and action: a review. *Current Opinion in Behavioral Sciences*, 42, 15-21. doi:10.1016/j.cobeha.2021.02.001
- Duan, R., Hepworth, K. J., Ormerod, K. J., & Canon, C. (2021). Promoting Concern for Climate Change: A Study of Wildfire Photographs Using Q Methodology. *Science Communication*, 43(5), 624-650. doi:10.1177/10755470211041689
- Ettinger, J., Walton, P., Painter, J., & DiBlasi, T. (2021). Climate of hope or doom and gloom? Testing the climate change hope vs. fear communications debate through online videos. *Climatic Change*, 164(1-2). doi:10.1007/s10584-021-02975-8
- Feldman, L., & Hart, P. S. (2021). Upping the ante? The effects of “emergency” and “crisis” framing in climate change news. *Climatic Change*, 169(1-2). doi:10.1007/s10584-021-03219-5

Kuller, M., Schoenholzer, K., & Lienert, J. (2021). Creating effective flood warnings: A framework from a critical review. *Journal of Hydrology*, 602. doi:10.1016/j.jhydrol.2021.126708

O'Neill, S. J., & Smith, N. (2013). Climate change and visual imagery. *WIREs Climate Change*, 5(1), 73-87. doi:10.1002/wcc.249

Reser, J. P., & Bradley, G. L. (2017). Fear Appeals in Climate Change Communication. In *Oxford Research Encyclopedia of Climate Science*.

Reviewer #2 (Remarks to the Author):

This paper presents the findings of a well-designed, timely and relevant study of the effects of imagery in communication about wildfire risks to homeowners. The combination of an expert survey, an online (laboratory) experiment, and a field experiment is commendable and provides a rich source of internally and externally valid results. The findings shed light on the interaction effects between use of imagery in risk communication and the actual reported risk in numerical terms on information-seeking behaviour by homeowners. The novel – and practically very relevant – finding is that risk imagery (here: a wildfire photo) tends to depress information-seeking behaviour the higher the actual reported risk is to a homeowner, compared to more “peaceful” imagery. This, of course, has implications for the usefulness of impact imagery in risk-related communication in times of a climate crisis (whether that’s public outreach campaigns, media coverage, corporate communication, etc.).

I’d like to see a version of this paper in print, since it contributes to the burgeoning field of visual effects research in climate change communication, and because of its usefulness to practitioners in climate risk communication. I have a few points of critique which relate to mainly three areas: (a) the role and significance of the expert survey; (b) the use of and reporting of confidence intervals for non-randomly selected data; and (c) the exact design of the online experiment. Plus, there are some minor editorial questions.

We thank the reviewer helpful comments that have improved how we describe and present our study. We have made revisions in response to the three areas noted here. In particular, we have done the following:

- a) motivated the role and significance of the expert survey through a discussion of the current controversy around using negative imagery
- b) removed confidence intervals for our non-randomly selected data; and
- c) clarified the design of the online experiment.

Please make sure that the geographical (and therefore political, cultural, economic, etc.) scope of your study is made explicit right from the start. It was the expression “wildfire-prone states” (line 69) that made me realize this is a U.S.-based study (actually, I already thought so because of the lack of specification in the abstract and opening paragraphs); please introduce the U.S. focus in the abstract.

We have specified the geographic focus of the study in the abstract and in the introduction (line 80-82):

“Here, we investigate practitioner assumptions about the use of negative imagery in risk communication and the effect of negative imagery on behavioral adaptation to a climate-exacerbated hazard in the United States.”

As regards the expert survey, I can see how it helps to illustrate the gap between practitioners’ perceptions and behavioural inclinations on the one hand and your experiments’ findings on the other. But beyond that it seems rather peripheral to and detached from the overall message of the paper.

With guidance from another reviewer, we have elaborated on the practical relevance of the practitioner survey in lines 67-79:

“Uncertainty about the effects of negative framing and emotions has led to controversy over their use in climate change communications.³⁴ The terms “climate crisis” and “climate emergency” are increasingly used by news outlets,³³ and scientists have called for greater use of images that make the negative effects of climate change more salient.³⁵ Yet theories and empirical work from other domains suggest this approach may backfire if is not accompanied by effective ways to take action.²⁶ Claims about behavioral effects of fear appeals and negative emotions in climate communications are often supported by evidence that is correlational, from experiments with small samples, or lacking domain specificity.^{13,34,36} As a result, there is an “evidence-advice” disparity about the use of negative imagery that may have produced confusion among practitioners.³⁶ Given the challenges associated with isolating the elements of any given image, experimental research should seek ecological validity aligned with the needs of practitioners.^{37”}

Relatedly, you report confidence intervals for your descriptive findings from the survey. Since these results are not based on a randomly drawn sample, the CI numbers don’t make a lot of sense. I’d also like to know a bit more about the message board you used for recruitment: Is it nationwide? How well does it match the population of “wildfire practitioners in the U.S.?” Do you have a notion of what kind of biases have been introduced by your recruitment and sampling strategy? What was the response rate?

Thank you for pointing out our error in reporting confidence intervals for this sample; we have dropped the intervals from the text.

We have elaborated on the description of the message board and its membership (lines 292-297):

“Practitioners were recruited through emails to contacts and postings on the Fire Adapted Communities Learning Network (FAC Net; <https://fireadaptednetwork.org/>) message board. FAC Net is a national network of wildfire professionals actively working on fire adaptation, including representatives from fire departments, conservation districts, nonprofits, and Firewise fire councils.”

We have also added text that states that we are unable to assess response rate from this study (lines 300-301):

“Because we do not know how many FAC Net members viewed the posting, we are unable to measure the response rate.”

Finally, we have added a discussion of the implications of our nonprobability sampling (lines 264-270):

“We note that the wildfire professionals who responded to our survey may not represent the broader population of wildfire practitioners. These individuals are part of an online community of fire professionals interested in building community resilience to wildfire. As such, these practitioners may be more informed about and experienced with wildfire outreach to homeowners. It is unclear whether or how their perceptions might differ from the broader population

of wildfire practitioners. However, this subpopulation is integral to building fire-adapted communities in the United States.⁵⁵”

I’m getting a bit confused about the design of the online experiment and the respective findings. Figure 1 only show two images and you often speak of the “flames photo” and the “status-quo photo” (singular) when describing the findings of the online experiment. But later on, you mention four photos and your supplementary material shows the other two images. You randomly showed four photos to your participants (Table S2), but I’m not sure what the reported findings of the online experiment in the main text are referring to. Are the calculations just based on the experimental exposure to the images in Figure 1 (i.e., the ones also used in the field experiment), or are they based on all four treatment conditions (but grouped into two)? I assume the later, but it’s just a bit confusing – especially when you write (on lines 259-260) that you wanted “to determine whether reactions to those two photos generalized to two similar photos.” Lines 137-183 suggest that you ran the tests twice.

We have rewritten the description of this study, which we hope better explains the design and how the findings are presented (lines 98-110):

“In an online experiment, homeowners ($n = 440$) in wildfire-prone states responded to flames and status quo photos, allowing us to identify relevant differences between the two images described above. Homeowners were told to imagine they live in an area at risk from wildfire and that they had received a photo postcard from the local fire department. They were randomly assigned to one of four photos: the flames photo, the status quo photo, or one of two similar photos as part of a sensitivity analysis (Supplementary Figure 1). While viewing the assigned photo, participants were asked about emotional reactions, probability assessments, behavioral intentions, and personal preferences. We tested for differences between the two photos used across the three studies (Figure 1), and report those results below. As a sensitivity analysis, we also tested whether differences are consistent with similar wildfire photos used in mainstream media and homeowner communications. Those results are provided in the Supplementary Information.”

We have also attempted to clarify the methods used across the different photos (lines 339-344):

“The goals of this study were (1) to measure differences between the two photos used in the field experiment; and (2) to determine whether reactions to those two photos generalized to two similar photos. Differences between the field experiment photos (Goal 1) were evaluated using t-tests with Benjamini-Hochberg corrections for multiple comparisons.⁶⁰ Generalizability of similarities and differences to the other two photos (Goal 2) was tested using one-way ANOVAs and post-hoc Tukey’s HSD.”

Lastly, you say (on lines 88-89) that “homeowners were randomly assigned based on their properties’ wildfire risk” (repeated on lines 290-291). That sounds contradictory. Were they assigned randomly or based on a pre-set criterion? I assume that you first grouped households based on their risk scores (e.g., everyone with a score of 700 was assigned to one group) and then within each risk-level-group the stimulus material was assigned randomly to households. That’d make sense. Can you please describe it so that nobody gets confused?

Thank you for pointing this out. We have clarified the text in the introduction (lines 122-124):

“Homeowners ($n = 5785$) were stratified based on their properties’ wildfire risk and randomly assigned to receive one of two versions of the postcard...”

The randomization approach is further described under Methods (lines 372-375):

“Assignment to treatment followed a randomized block design, in which homeowners were randomly and evenly distributed between the treatment and control groups according to their wildfire risk rating (Low to Extreme), which was classified by wildfire experts using wildfire risk scores.”

The online questionnaire could be added to the supplementary material

The online questionnaire has been added to the supplementary information. This is noted in line 338:

“The full questionnaire is provided in Supplementary Methods.”

The following are some minor editorial comments:

- line 25: maybe explain in a few words who these practitioners are (as you do later in the manuscript);

We have rephrased and added text to specify what we mean by practitioners:

“Practitioners—employees of government agencies and nonprofits working to encourage and support individual actions to mitigate and adapt to these hazards—must decide whether to use such negative imagery in their risk communications.”

- line 76: where exactly did you find these photos and why did you choose them as experimental stimuli?

This information is provided under Methods (lines 326-329):

“Two photos represented the status quo because they had been used by fire departments to communicate with homeowners about wildfire risk. Two photos represented flames imagery because they showed burning houses and had appeared in media outlets along with stories about wildfire destruction.”

- line 94: you could be more specific and write “unique access codes for every household, which needed to be entered when visiting the website” or something similar;

We have improved this statement as suggested (lines 127-129):

“Webpage visits were tracked at the homeowner level using access codes unique to each household, which were entered to view a property’s webpage.”

- line 116: should read “no agreement;”

We thank the reviewer for catching this error – it has been fixed.

- line 147: some readers might wonder what you mean by “parcel” here; you could also list two or three owner characteristics for illustrative purposes;

We have added examples as suggested (lines 180-181):

“The regression model includes adjustments for parcel and owner characteristics, including property acreage and value.”

- line 166: “did induce more negative and less positive emotions” is correct but sounds a bit clunky.

We have attempted to improve this sentence (lines 203-204):

“The flames photo induced stronger negative emotions and weaker positive emotions than the status quo photo among homeowners in the online experiment.”

Dr Antal Wozniak (University of Liverpool, Department of Communication and Media)

Reviewer #3 (Remarks to the Author):

Thank you for the opportunity to review “The effects of wildfire imagery in risk communications”. In this manuscript, the authors explore evocative negative imagery and wildfire risk communication. The authors survey wildfire practitioners on their attitudes about the use of negative images, and conduct two surveys of members of the public to assess how negative images influences willingness to engage in information seeking about protective behavior. Overall, I think the authors did a good job and the study will be particularly useful for practitioners looking to design risk communication campaigns informed by best available science. The statistical analyses are appropriate for the research questions proposed by the authors and the nature of the data collected. I have one issue with interpretation of the results, as described in greater detail below. The authors provide suitable detail to reproduce this study, although they may want to consider including their survey items as supplementary material in the appendix.

I enjoyed reading the paper and I hope the authors find these comments helpful.

We thank the reviewer for helpful comments that have strengthened how we ground our research in the existing literature and discuss our findings. We have also included our survey items under Supplementary Methods in the Supplementary Information.

Major Comments

1. Lines 98 – 99: “Theory also predicts...” Practitioners and academic readers of Communications Earth & Environment may be unfamiliar with many of the theories of risk communication frequently utilized in the homeowner preparedness/wildfire risk communication literature. The authors may want to consider whether they should name and define some theories and models of risk communication and risk appraisal (e.g., Risk Information Seeking and Processing Model, Extended Parallel Process Model, Protection Motivation Theory, etc.) both here and more generally in the introduction and discussion. This may help readers situate the results in a larger theoretical context.

This is a great point. We have extended the theoretical grounding of our study in several ways. First, by elaborating on Protection Motivation Theory and the Extended Parallel Process Model in the Introduction (lines 46-66):

“Fear appeals, which highlight the severity and saliency of threats, are a common strategy in risk communications.^{22,23} According to Protection Motivation Theory, the effects of fear appeals depend on cognitive appraisal processes, including perceptions of severity, vulnerability, and response efficacy and costs.²⁴ Adaptive responses to fear appeals are guided by coping appraisal, which includes ability to address the threat (*self-efficacy*), effectiveness of action (*response efficacy*), and response costs. When perceptions of efficacy are high, fear appeals can increase protective behaviors.^{22,25} This pathway has been established empirically in the health communications literature, but few empirical studies have causally linked appeals and action in climate and environmental domains.²⁶

Conversely, highlighting the negative consequences of climate risks could dissuade action. Maladaptive responses to fear appeals (e.g., avoidance or denial) may occur when there are intrinsic or extrinsic rewards for avoiding the threat,²⁴

or when perceptions of high severity and vulnerability are not accompanied by efficacy (according to the Extended Parallel Process Model).²⁷ While imagery of climate impacts increases threat salience, it rarely elicits self-efficacy.²⁸ Without self-efficacy, negative emotions can generate hopelessness.²⁹ Undesirable emotions or information could prompt willful ignorance, avoidance, or even reactance among those who do not want to be scared into submission.³⁰ There is also a risk of compassion fatigue, in which emotional appeals lead to exhaustion or desensitization,³¹ or skepticism, if emotional appeals or strong terminology reduce a message's credibility.^{32,33}

As well as interpreting our results in the context of this theory in the Discussion (lines 225-243):

“The importance of perceived efficacy in response to the flames imagery is consistent with both the wildfire context and the broader theoretical literature on fear appeals and negative affect in climate communications.^{27,47} Wildfire risk to private property includes factors that are beyond a homeowner's control.⁴⁸ A densely forested neighboring property and a steep hillside can greatly increase a property's wildfire risk, yet there is little the homeowner can do to change those factors. In this case, perceived efficacy is likely to be low and the homeowner will attempt to reduce fear rather than danger.²⁷ Even factors within a homeowner's control, such as roof material or defensible space, can be costly to change and even cost-prohibitive for those who are resource-constrained.⁴⁹ The response costs or “resource-related attributes” of mitigation action (time, financial cost, effort, and cooperation requirements) may have induced a maladaptive response.^{24,47} That the flames imagery increased webpage visitation among low-risk property owners but reduced visitation among high-risk property owners may precisely demonstrate the “tradeoff between salience and efficacy” of fear appeals,³⁷ and the “threat-by-efficacy” interaction explained by the Extended Parallel Process Model.²⁷ Moreover, while climate change may be perceived as so large and unsolvable as to inhibit self-efficacy and response efficacy,²⁶ highlighting even more localized climate-related threats (e.g., to one's own property) may be paralyzing enough to induce maladaptive responses.”

2. The authors find that wildfire practitioners typically believe the evocative negative imagery is more effective at motivating information-seeking behavior, the effect of negative imagery depends on the characteristics of the individual. The authors recommend researchers and practitioners collaborate to improve risk communication to encourage mitigation behavior. I think the paper would be strengthened by a discussion of management implications to advise practitioners on when and how to use negative imagery, given the literature the authors reviewed in the introduction, the results of this study, and the potential theoretical justifications the authors explore in the discussion.

We have added a paragraph to the Discussion that advises practitioners about the use of negative imagery (lines 271-279):

“The key result of the field experiment for these and other practitioners is that negative imagery and fear appeals can backfire among those who are at greatest risk to climate-related hazards. Related theory suggests the importance of

providing feasible options for risk-reduction to avoid this negative response.^{27,48} While the postcard in the field experiment advertised a website with “specific information on how to reduce wildfire risk,” this text may not have been enough to attenuate the resource-related or efficacy concerns of high-risk homeowners. Identifying and addressing those concerns may be key to using negative imagery in risk communications, which may indeed have attention-grabbing benefits.”

3. Some parts of the online experiment are unclear. From my understanding, the authors asked respondents to imagine they lived in a home at risk of wildfire. Were respondents asked how likely they would be to visit a personalized website for their imagined residence or for their actual residence? Can the authors provide some background on why they chose to have respondents imagine they owned a home at risk of wildfire rather than have respondents answer from the perspective of their current residence?

We have added text to explain the Methods used in the online experiment (lines 332-336):

“Study design and questions were modeled after similar studies on disaster risk perceptions and emotional reactions to imagery.^{11,19,58,59} Specifically, participants were asked to imagine they owned a home in an area at risk from wildfire to make the hazard relevant and provide a specific, consistent risk measure for all participants.¹⁹”

We have also added the questionnaire used in the online experiment to the supplementary information to provide more clarity to readers on how that study was conducted.

4. I’m not sure I follow some of the interpretation of the field experiment in the discussion. Lines 151 – 157, the authors mention twice that seeing the flames photo makes high-risk respondents less likely to visit the webpage. I would clarify that for high risk respondents, the negative emotions photo made them less likely to visit the webpage than high risk respondents who received the neutral photo. Based on Figure 4, it looks like respondents at high fire risk were more likely to view the webpage than low risk respondents regardless of treatment. However, the authors seem to imply in lines 170 – 180 that those who were highly susceptible to wildfire risk may be more likely to engage in fear control responses like avoidance and willful ignorance (even though across both conditions, the higher your susceptibility, the more likely you were to seek out information.

Thank you for pointing this out. We have clarified the text in the Results (lines 186-191):

“But for every 100-point increase in risk score—which ranged from zero to 1000 points—the flames photo decreased web visits by 2.3 percentage points compared to the status quo photo ($p < 0.01$; 95% CI [-3.8, -0.8]; Figure 3). For a homeowner whose risk score is one standard deviation higher than average, the flames photo reduced webpage visits by 3.5 percentage points compared to the status quo baseline.”

As well as in the Discussion:

“In fact, among higher-risk properties—where mitigation is most important—the flames photo reduced homeowner engagement compared to a status quo landscape image.” (lines 197-199)

“If these emotional effects replicated in the field, perhaps fear and worry attracted the attention of all who received the flames photo, but those emotions interacted with the personal wildfire risk information on the postcard. Fear appeals tend to work best when accompanied by high perceived efficacy, or effective and feasible options for responding to the threat.⁴⁴ Receiving the flames photo and then learning one’s property was low risk may have encouraged a sense of efficacy, enabling protective information-seeking behavior.⁴⁵ Conversely, homeowners who received the flames photo and then learned their property was high risk were faced with more challenging coping strategies. This combination of high susceptibility to and low efficacy to avert a threat can lead people to engage in fear control responses.²⁷” (lines 323-333)

Methods and Figures

1. How did the authors ensure their respondents were homeowners? In the Ashland, OR, sample, how did the authors treat residences where the occupants were renters? Did the authors exclude multi-family structures like apartments in addition to non-residential properties?

The sample characteristics are described under Methods (lines 354-361):

“The sample frame included every parcel within the city limits of Ashland, Oregon (N = 6,400). Property owners were identified using the Jackson County Property Data list (<https://web.jacksoncounty.org/pdo/>). Non-residential properties were excluded from the list. Owners of multiple properties were identified using a name-matching process, which resulted in 5,785 unique postcard recipients. Homeowners were also identified as part-time residents if their mailing address was outside the city. A mailing list was constructed from the Property Data list and matched to parcel-level wildfire risk assessment data collected by the local wildfire division.”

2. Did the authors send multiple waves of postcards in the field experiment, or just one?

There was only one postcard mailing, which is described in the Methods (lines 362-363):

“Postcards were mailed to all homeowners in Ashland, Oregon by the wildfire division of the local fire department in July 2020.”

3. Were any of the parcel characteristics or ownership characteristics related to information-seeking behavior?

While this is an interesting question, it is beyond the scope of our study’s investigation. However, we have added an additional table to the Supplementary Information that describes the parcel and ownership characteristics of the property owners that visited their webpages overall and by treatment (Supplementary Table 3). Curious readers are able to compare these descriptive statistics with those provided for the full sample (Supplementary Table 1).

4. Table S1 should include the descriptive statistics for the sub-sample who visited the webpage. I recommend replacing the Flames and Status Quo columns with two columns for the Flames subsample who visited the page and the Status Quo subsample who visited the page. I recommend Table S1 be brought into the main manuscript.

Supplementary Table 3 has been added to the Supplementary Information and provides the subgroup characteristics of those who visited the webpage. We have left Supplementary Table 1 (formerly Table S1) as it was because it provides important information about the overall sample and balance across treatments.

5. Figure 2: I recommend the authors replace Figure 2 with Table S2.

We have declined to make this change because Figure 2 highlights the differences between the photos used in the other two studies described in the manuscript. Table S2 (now Supplementary Table 2) includes additional information about two other photos to test for consistency; this information is not central to the arc of the paper and may confuse the reader.

6. Figure 3: I recommend the authors replace Figure 3 with Table S3.

Thank you for this suggestion; we have replaced Figure 3 with Table S3 (now Table 2).

Minor Comments

1. The authors include the reference number superscript inside the period rather than outside throughout the manuscript.

We have corrected this throughout the manuscript.

2. In the supplementary materials, the figures and tables are not listed in the order they appear in the manuscript. For example, the authors reference Figure S3 and S4 on line 138, well before Figures S1 and S2 are references on line 283.

3. Extremely minor, but consider adding in the title of Figure S1 that the example address is the Ashland Fire and Rescue and not a home address of a potential respondent.

We have added this clarification to the title of Figure S1 (now Supplementary Figure 2).

19th May 22

Dear Dr Byerly Flint,

Your manuscript titled "The effects of wildfire imagery in risk communications" has now been seen by our reviewers, whose comments appear below. In light of their advice I am delighted to say that we are happy, in principle, to publish a suitably revised version in Communications Earth & Environment under the open access CC BY license (Creative Commons Attribution v4.0 International License).

We therefore invite you to revise your paper one last time to address the remaining concerns of our reviewers. At the same time we ask that you edit your manuscript to comply with our format requirements and to maximise the accessibility and therefore the impact of your work.

EDITORIAL REQUESTS:

Please review our specific editorial comments and requests regarding your manuscript in the attached "Editorial Requests Table". Please outline your response to each request in the right hand column. Please upload the completed table with your manuscript files.

SUBMISSION INFORMATION:

OPEN ACCESS:

Communications Earth & Environment is a fully open access journal. Articles are made freely accessible on publication under a [CC BY license](http://creativecommons.org/licenses/by/4.0) (Creative Commons Attribution 4.0 International License). This license allows maximum dissemination and re-use of open access materials and is preferred by many research funding bodies.

For further information about article processing charges, open access funding, and advice and support from Nature Research, please visit <https://www.nature.com/commsenv/article-processing-charges>

At acceptance, you will be provided with instructions for completing this CC BY license on behalf of all authors. This grants us the necessary permissions to publish your paper. Additionally, you will be asked to declare that all required third party permissions have been obtained, and to provide billing information in order to pay the article-processing charge (APC).

[link redacted]

Best regards,

Joe Aslin

Senior Editor,
Communications Earth & Environment
<https://www.nature.com/commsenv/>
Twitter: @CommsEarth

REVIEWERS' COMMENTS:

Reviewer #1 (Remarks to the Author):

I am satisfied with the revision that the authors have made to my queries, and I think this paper represents a nice addition to the literature.

As a final comment, I do point to the following argument made by reviewer 3:

'I'm not sure I follow some of the interpretation of the field experiment in the discussion. Lines 151 – 157, the authors mention twice that seeing the flames photo makes high-risk respondents less likely to visit the webpage. I would clarify that for high risk respondents, the negative emotions photo made them less likely to visit the webpage than high risk respondents who received the neutral photo. Based on Figure 4, it looks like respondents at high fire risk were more likely to view the webpage than low risk respondents regardless of treatment. However, the authors seem to imply in lines 170 – 180 that those who were highly susceptible to wildfire risk may be more likely to engage in fear control responses like avoidance and willful ignorance (even though across both conditions, the higher your susceptibility, the more likely you were to seek out information.'

I agree with this reviewer that it is important to clarify in the main text that 'high fire risk were more likely to view the webpage than low risk respondents regardless of treatment.' I am missing this finding currently in the main results section, but I think this information is important to adequately interpret the results. I would therefore suggest still adding this to the results section of Study 3.

Reviewer #2 (Remarks to the Author):

I think the authors have responded to the reviewers' points of critique and recommendations very well and comprehensively. The manuscript is in really good shape now. I only have very minor, purely editorial suggestions:

“Contrariety” (line 3) sounds unusual to my ears, but if you think it is the most accurate word please leave it.

Maybe write “conflicting assumptions” in line 4.

When describing empirical findings in the introduction, please stick to present tense throughout, e.g., on line 27, instead of “have increased” you could say “have been shown to increase” or “can increase,” and it could be “imagery of flooding leads” on line 39.

Is “wellspring of action” on line 30 a direct quote from one of the readings? In which case the citation should be added right after it.

An “it” is missing as the fourth word on line 69.

On line 101, should it not say “after” instead of “while” (or were the participants actually looking at the photo whilst filling out the questionnaire)?

On line 136, should it not read “effective in” instead of “effective at?”

It should read “deemed less personally relevant” (or something similar) on line 165.

I'm looking forward to seeing the paper in print!

Reviewer #3 (Remarks to the Author):

The authors have done a good job revising their paper in response to reviewer comments. I have three minor comments. Other than these minor comments, I believe the paper is acceptable for publication. Good job!

1. Line 130: “Theory also predicts that higher risk leads to information insufficiency...” I would state which theory/theories predict this.
2. Line 132. Consider adding a link or other way to find the pre-registration here.
3. For the parentheticals with the Mdiff statistics (e.g., Lines 152 – 166), consider replacing the confidence interval with the t-statistic and p-value from the difference of means test, with correction for multiple tests where relevant.

REVIEWERS' COMMENTS:

Reviewer #1 (Remarks to the Author):

I am satisfied with the revision that the authors have made to my queries, and I think this paper represents a nice addition to the literature.

As a final comment, I do point to the following argument made by reviewer 3:

'I'm not sure I follow some of the interpretation of the field experiment in the discussion. Lines 151 – 157, the authors mention twice that seeing the flames photo makes high-risk respondents less likely to visit the webpage. I would clarify that for high risk respondents, the negative emotions photo made them less likely to visit the webpage than high risk respondents who received the neutral photo. Based on Figure 4, it looks like respondents at high fire risk were more likely to view the webpage than low risk respondents regardless of treatment. However, the authors seem to imply in lines 170 – 180 that those who were highly susceptible to wildfire risk may be more likely to engage in fear control responses like avoidance and willful ignorance (even though across both conditions, the higher your susceptibility, the more likely you were to seek out information.'

I agree with this reviewer that it is important to clarify in the main text that 'high fire risk were more likely to view the webpage than low risk respondents regardless of treatment.' I am missing this finding currently in the main results section, but I think this information is important to adequately interpret the results. I would therefore suggest still adding this to the results section of Study 3.

We have added the following sentence to the Results section of Study 3:

“Regardless of the photo on their postcard, homeowners with higher-risk properties were more likely than those with low-risk properties to visit their personalized risk webpages.”

Reviewer #2 (Remarks to the Author):

I think the authors have responded to the reviewers' points of critique and recommendations very well and comprehensively. The manuscript is in really good shape now. I only have very minor, purely editorial suggestions:

“Contrariety” (line 3) sounds unusual to my ears, but if you think it is the most accurate word please leave it.

We have dropped this word from the Abstract.

Maybe write “conflicting assumptions” in line 4.

We have made this revision.

When describing empirical findings in the introduction, please stick to present tense throughout, e.g., on line 27, instead of “have increased” you could say “have been shown to increase” or “can increase,” and it could be “imagery of flooding leads” on line 39.

We have updated the text throughout the Introduction to be present tense.

Is “wellspring of action” on line 30 a direct quote from one of the readings? In which case the citation should be added right after it.

We have moved the citation to follow the quotation.

An “it” is missing as the fourth word on line 69.

We have added this missing word.

On line 101, should it not say “after” instead of “while” (or were the participants actually looking at the photo whilst filling out the questionnaire)?

We have made this change.

On line 136, should it not read “effective in” instead of “effective at?”

We have made this change.

It should read “deemed less personally relevant” (or something similar) on line 165.

We have made this change.

I'm looking forward to seeing the paper in print!

Reviewer #3 (Remarks to the Author):

The authors have done a good job revising their paper in response to reviewer comments. I have three minor comments. Other than these minor comments, I believe the paper is acceptable for publication. Good job!

1. Line 130: “Theory also predicts that higher risk leads to information insufficiency...” I would state which theory/theories predict this.

We have clarified the theory to which we are referring:

“The Risk Information Seeking and Processing Model also predicts that higher risk leads to information insufficiency...”

2. Line 132. Consider adding a link or other way to find the pre-registration here.

We have added a link to the pre-registration.

3. For the parentheticals with the Mdiff statistics (e.g., Lines 152 – 166), consider replacing the confidence interval with the t-statistic and p-value from the difference of means test, with correction for multiple tests where relevant.

We have declined to make this change but we have added an additional table to the Supplementary Material that provides this information. We refer to this table in the main text:

“Test statistics and p-values are reported in Supplementary Table 2.”